



# Application of Generalized - Aurora Computed Tomography to the EISCAT_3D project

Yoshimasa Tanaka[1,2,3], Yasunobu Ogawa[1,2,3], Akira Kadokura[1,2,3], Takehiko Aso[1], Björn Gustavsson[4],
Urban Brändström[5], Tima Sergienko[5], Genta Ueno[6], Satoko Saita[7]

[1]National Institute of Polar Research, Tachikawa, 190-8518, Japan
[2]Polar Environment Data Science Center, ROIS-DS, Tachikawa, 190-0014, Japan
[3]Graduate Institute for Advanced Studies, SOKENDAI, Tachikawa, 190-8518, Japan
[4]UiT The Arctic University of Norway, Tromsø, 6050, Norway
[5]Swedish Institute of Space Physics, Kiruna, 812, Sweden
[6]The Institute of Statistical Mathematics, Tachikawa, 190-8562, Japan
[7]Kitakyushu National College of Technology, Kitakyushu, 802-0985 Japan

*Correspondence to*: Yoshimasa Tanaka (ytanaka@nipr.ac.jp)

**Abstract.** EISCAT_3D is a project to build a multiple-site phased-array incoherent scatter radar system in northern Fenno-
Scandinavia. We demonstrate via numerical simulation how useful monochromatic images taken by a multi-point imager network are for auroral research in the EISCAT_3D project. We apply the generalized-aurora computed tomography (G-ACT) method to modelled observational data from real instruments, such as the Auroral Large Imaging System (ALIS) and the EISCAT_3D radar. The G-ACT is a method for reconstructing the three-dimensional (3D) distribution of auroral emissions and ionospheric electron density (corresponding to the horizontal two-dimensional (2D) distribution of energy
spectra of precipitating electrons) from multi-instrument data. It is assumed that the EISCAT_3D radar scans an area of 0.8° in geographic latitude and 3° in longitude at an altitude of 130 km with 10×10 beams from the radar core site at Skibotn (69.35°N, 20.37°E). Two neighboring discrete arcs are assumed to appear in the observation region of the EISCAT_3D radar. The reconstruction results from the G-ACT are compared with those from the normal ACT as well as the ionospheric electron density from the radar. It is found that the G-ACT can interpolate the ionospheric electron density at a much higher
spatial resolution than that observed by the EISCAT_3D radar. Furthermore, the multiple arcs reconstructed by the G-ACT are more precise than those by the ACT. Even when the ACT reconstruction is difficult due to the unsuitable locations of the imager sites relative to the discrete arcs and/or a small number of available images, the G-ACT allows us to obtain better reconstruction results.

## 1 Introduction

EISCAT_3D is a multi-point phased array incoherent scattering radar system under construction in northern Fenno-Scandinavia as of November 2023 and is expected to be operational in winter 2023. The EISCAT_3D radar will be able to measure the three-dimensional (3D) distribution of ionospheric parameters, such as the electron density, electron temperature,





ion temperature, and ion Doppler velocity, at a resolution that is more than 10 times higher than that of the existing EISCAT

radar. Thus, it is expected to provide new insights into various science topics pertaining to auroral physics, ionospheric

physics, magnetosphere-ionosphere coupling, and so on (McCrea et al., 2015; Wannberg et al., 2010).

The height distribution of the ionospheric electron density in the auroral region, which is related to the energy distribution of

auroral precipitating electrons, is essential for clarifying if the precipitating electrons experienced acceleration and to

determine their place of origin. In addition, we can estimate the ionospheric conductivity from the electron density by using

empirical models (e.g., the Mass Spectrometer and Incoherent Scatter (MSIS) atmosphere model and the International

Geomagnetic Reference Field (IGRF) model (Hedin, 1991; Alken et al., 2021). It is well known that the spatial distribution

of ionospheric conductivity plays an essential role in the magnetosphere-ionosphere coupling process (e.g., Ellis and

Southwood, 1983; Glaβmeier, 1984; Itonaga and Kitamura, 1988). It may be possible to deduce a 3D current system from

the ionospheric conductivity distribution by using magnetic field data from a ground-based magnetometer array and/or

ionospheric electric field data from radars (Kamide et al., 1981; Vanhamäki and Amm, 2007).

On the other hand, it is useful to utilize optical imaging observations to study auroral dynamics. Radars generally have a high

range resolution; however, scanning a particular area with multiple beams is time-consuming. In contrast, an optical imager

has a high angular resolution. It can measure angular distributions at a higher temporal resolution than a radar, even though

they can detect only the integrated luminosity along the line of sight. In other words, the radar and optical imager are

complementary. Therefore, it is essential to use image data with radar data effectively. Figure 1 shows a schematic

illustration of the relationship between the radar and imager observations.

There are some ground-based imager networks in northern Fenno-Scandinavia, for example, the Aurora Large Imaging

System (ALIS) (Brändström, 2003), the all-sky cameras in the Magnetometers - Ionospheric Radars - All-sky Cameras Large

Experiment (MIRACLE) (Syrjäsuo, 2001), and Watec Monochromatic Imager (WMI) (Ogawa et al., 2020). In particular, the

ALIS was designed to obtain the 3D distribution of the optical emissions in the mesosphere, thermosphere, and ionosphere,

and was recently developed into ALIS_4D (https://alis4d.irf.se/). By applying the auroral computed tomography (ACT)

technique to the monochromatic images taken at some ALIS stations, it is possible to retrieve the 3D distribution of auroras

that have a horizontal scale of several tens to hundreds of kilometers (Aso et al., 1998; Gustavsson, 1998; Gustavsson et al.,

2001; Simon Wedlund et al., 2013). Conversely, the inverse problem of the ACT is ill-posed and ill-conditioned, because the

optical image data correspond to the emission intensity integrated along the line of sight and only a few images are usually

available. Thus, assumptions need to be made to solve the inverse problem, which often makes it somewhat difficult to

interpret the results of the data analysis from a physical point of view.

We have extended the ACT to the generalized ACT (G-ACT) (Aso et al., 2008; Tanaka et al., 2011). This method can

reconstruct the spatial and energy distributions of auroral precipitating electrons by using multi-instrument data, such as the

ionospheric electron density from an incoherent scatter radar, the cosmic noise absorption (CNA) from an imaging riometer,

and optical monochromatic images. Tanaka et al. (2011) demonstrated via numerical simulation that only the reconstruction





from auroral images can be improved by the G-ACT by using the height profile of the electron density from the EISCAT radar.

In the current study, we investigated how effective the combination of the EISCAT_3D radar and the monochromatic imager network is for auroral research by conducting a simulation. We apply the G-ACT method to the modelled observational data,

i.e., electron density from the EISCAT_3D radar and the multiple monochromatic images from the ALIS and compare the reconstruction results with those obtained by the normal ACT and the radar's electron density data. We selected the ALIS (not ALIS_4D) as the monochromatic imager network for this simulation study, because the monochromatic images from the ALIS can be used for both the normal ACT analysis and the G-ACT analysis. It is possible to compare the auroral 3D distributions reconstructed by the two methods in the same region because one of the ALIS stations is located in Skibotn

(69.35°N, 20.37°E), Norway, which is the core site of the EISCAT_3D radar, and the field-of-view (FOV) of the ALIS imagers covers the radar observation region.

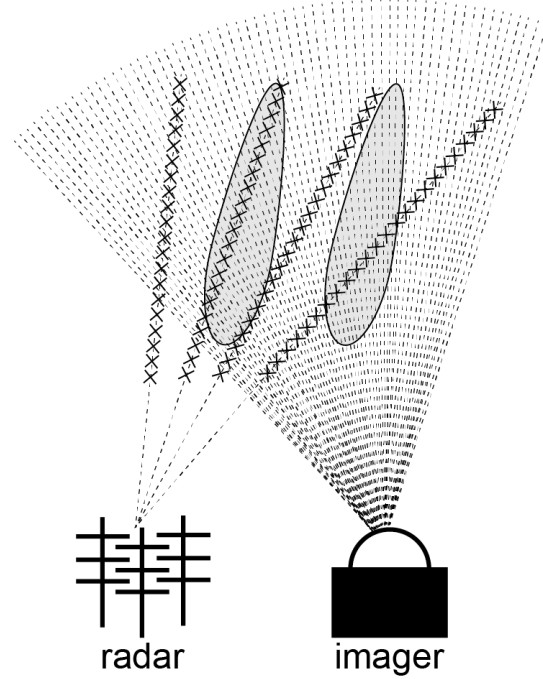

**Figure 1: Schematic illustration of the relationship between the radar and optical imager observations.**

## 2 Forward analysis

### 2.1 Observatories and instruments

Figure 2 shows the locations of the stations used in this simulation study and each instrument's field of views (FOVs) at an altitude of 130 km. Blue quadrangles and red crosses correspond to the FOVs of the ALIS imagers and the beam position of the EISCAT_3D radar, respectively. The core site of the EISCAT_3D radar is located at Skibotn, Norway.





It was assumed that the EISCAT_3D radar scanned an area of geographic latitude from 68.6°N to 69.4°N and longitude from

18.767°E to 21.767°E at an altitude of 130 km with 10×10 beams, which corresponds to a spatial resolution of 0.08° (about
8.9 km) in latitude and 0.3° (about 12 km) in longitude. It was also assumed that the electron density was detected at
altitudes between 90 and 170 km. The dashed-dotted line indicates the region where the reconstruction results were
evaluated, which is the same as the dashed-dotted line in Figure 3a.

Each ALIS station has a sensitive high-resolution (1024×1024 pixels) unintensified monochromatic CCD imager with a six-

position filter-wheel for narrow-band interference filters (427.8, 557.7, 630.0, and 844.6 nm) (Brändström, 2003). The FOV
of each imager is about 50 to 90°. It was assumed that the viewing direction was set to the south of Skibotn and the filter was
fixed to the $N_2^+$ 1st negative band (427.8 nm) for all stations. We postulated that the image size was reduced to 256×256
after 4×4 pixel binning.

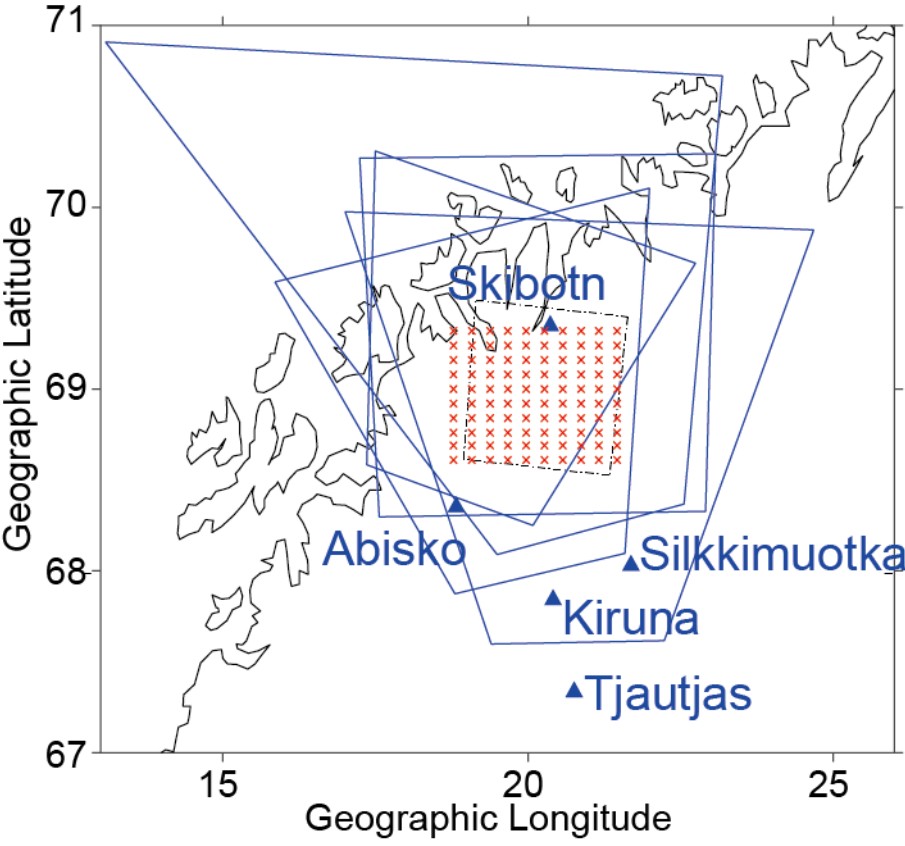

**Figure 2: Locations of the stations used in this study and the field of views (FOVs) of the ALIS imagers and EISCAT_3D radar at
an altitude of 130 km. It was assumed that the viewing direction of the imagers was set to the south of Skibotn and the radar
scanned an area of geographic latitude from 68.6°N to 69.4°N and longitude from 18.767°E to 21.767°E with 10×10 beams.**



## 2.2 Distribution of incident auroral electrons

Figure 3a indicates the horizontal distribution of the total energy flux ($Q_0$) of the auroral precipitating electrons that was
assumed for the forward analysis. It was also assumed that two neighboring discrete arcs appeared over the southern sky of
Skibotn. In this simulation, an oblique coordinate system was adopted with an origin at Skibotn, with the x-axis pointing in
the geomagnetic southward direction, the y-axis in the eastward direction, and the z-axis anti-parallel to the geomagnetic
field (c.f., Figure 2 of Tanaka et al., 2011). The inclination and declination angles of the geomagnetic field were 78° and 6°,
respectively. The calculation ranges were -40 to 100 km, -150 to 150 km, and 90 to 190 km for the $x$, $y$, and $z$ directions,
respectively. The spatial mesh size ($\Delta_x$, $\Delta_y$, $\Delta_z$) were 1 km, 2 km, and 2 km for the $x$, $y$, and $z$ directions, respectively.

The discrete arcs were assumed to have a sinusoidal shape in the $y$ direction and a Gaussian shape in the $x$ direction. The
distance between the two arcs was 20 km. The area surrounded by a thick line is the reconstruction region used for the
inverse analysis in Section 3. The dash-dotted line indicates the region where the reconstruction results are evaluated in
Section 4.



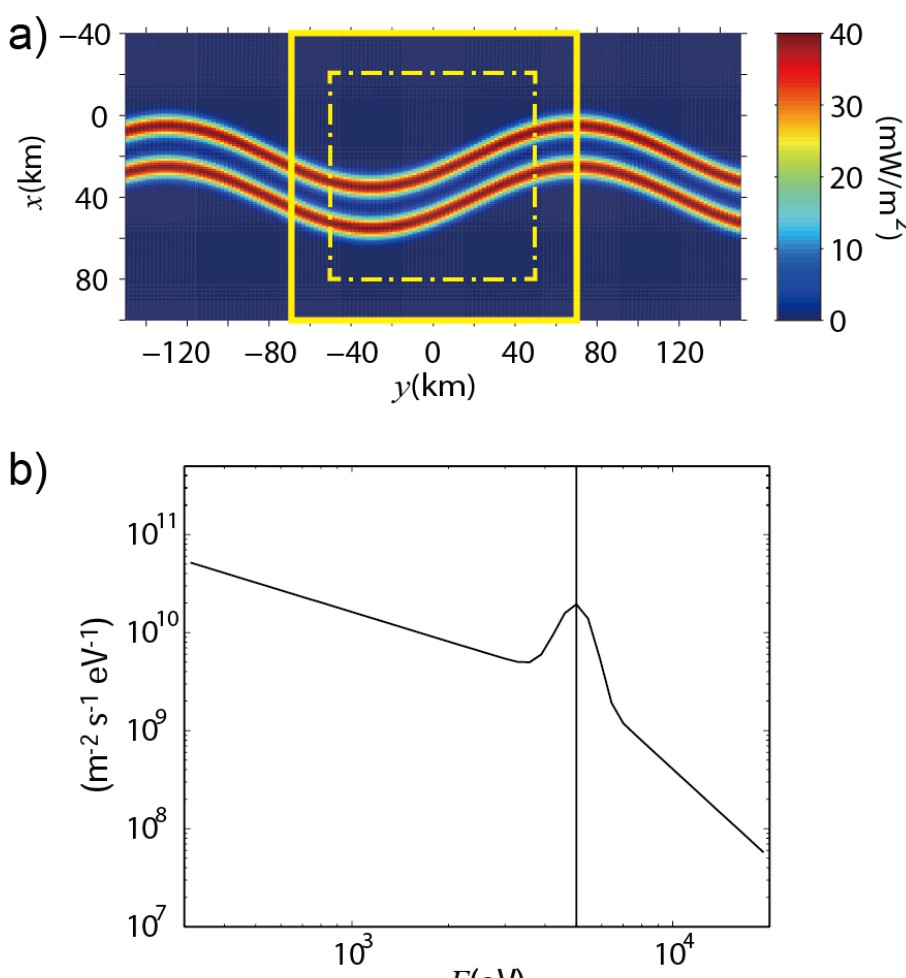

**Figure 3: (a) Horizontal distribution of the total energy flux ($Q_0$) of the incident auroral electrons. The top and right correspond to the geomagnetic northward and geomagnetic eastward directions, respectively. The thick line and dash-dotted line indicate the region used for the inverse analysis and the region where the reconstruction results were evaluated, respectively. (b) Energy distribution of the incident electrons at the peak location of the discrete arcs.**



Figure 3b shows the energy distribution of the differential number flux of the precipitating electrons at the peak location of the arcs. The energy spectrum is represented by the sum of the Gaussian distribution ($f_g(E) = A_g \exp\{-(E-E_0)^2 / W^2\}$) and two power-law distributions for the low-energy tail ($f_{pl}(E) = A_{pl}(E/E_0)^{-a}$, $E \leq E_0$) and high-energy tail ($f_{ph}(E) = A_{ph}(E/E_0)^{-b}$, $E \geq E_0$), which has been introduced by Strickland et al. (1993)

as a typical spectrum of discrete auroras. $E_0$, $W$, $a$, and $b$ were set to 5 keV, 0.15*$E_0$, 1.0, and 3.0, respectively, for the entire





simulation region. The energy range for the calculation was between 0.3 and 20 keV and divided logarithmically into 50 intervals.

Since the main purpose of this paper is to compare the results from the two analysis methods, ACT and G-ACT, we assumed a rough but typical auroral shape, size, and energy distribution. Actual auroras have a variety of shapes, including multiple
arcs, structured shapes, patchy shapes, and very thin arcs less than 100 m thick. However, it is difficult to examine such a large number of auroral types in this paper due to a lack of space. If one wants to evaluate the accuracy of the tomographic analysis results for real auroras, simulations should be performed using modelled auroras that resemble them (e.g., Fukizawa et al. 2022).

**2.3 Calculation of modelled data set**

The formulation of our method is based on that used by Janhunen (2001). The forward problem was solved by using the distribution of the incident electrons described in Subsection 2.2. The height profile of volume emission rate $\mathbf{L}_{x_1,y_1}(z)$ along the field line at a certain horizontal coordinate $(x_1, y_1)$ is calculated by

$$\mathbf{L}_{x_1,y_1}(z) = \mathbf{m}_1 \mathbf{f}_{x_1,y_1}(E), \tag{1}$$

where $\mathbf{f}_{x_1,y_1}(E)$ is the energy distribution of differential number flux of incident electrons at the top of the ionosphere, $(x_1,$
$y_1, z_{max})$ and $\mathbf{m}_1$ is a matrix operator for calculating $\mathbf{L}_{x_1,y_1}(z)$ from $\mathbf{f}_{x_1,y_1}(E)$. We adopted Rees' model (Rees, 1993) to obtain the energy deposition rate to the atmosphere from the differential flux and the method proposed by Sergienko and Ivanov (1993) to calculate the 427.8-nm volume emission rate from the energy deposition rate. The elements of $\mathbf{m}_1$ is the function of the atmospheric parameters, which were calculated by using the MSIS-90 atmosphere model (Hedin, 1991). The derivation of $\mathbf{m}_1$ is described in detail in the Appendix of Tanaka et al. (2011).

Assuming that $\mathbf{m}_1$ is independent of $x$ and $y$, Eq. (1) can be expanded in the $x$ and $y$ directions as follows:

$$\begin{pmatrix} \mathbf{L}_{1,1} \\ \mathbf{L}_{1,2} \\ \vdots \\ \mathbf{L}_{n_x,n_y} \end{pmatrix} = \begin{pmatrix} \mathbf{m}_1 & 0 & \cdots & 0 \\ 0 & \mathbf{m}_1 & \ddots & \vdots \\ \vdots & \ddots & \ddots & 0 \\ 0 & \cdots & 0 & \mathbf{m}_1 \end{pmatrix} \begin{pmatrix} \mathbf{f}_{1,1} \\ \mathbf{f}_{1,2} \\ \vdots \\ \mathbf{f}_{n_x,n_y} \end{pmatrix} \tag{2}$$

$$\mathbf{L} = \mathbf{M}_1 \mathbf{f}. \tag{3}$$

In (4), $\mathbf{f}$ is a function of $(x, y, E)$ and has a length of $n = n_E \times n_x \times n_y$, and $\mathbf{L}$ is a function of $(x, y, z)$ and has a length of

$m = n_z \times n_x \times n_y$. $\mathbf{M}_1$ is a large sparse matrix whose size is $m \times n$.





In a similar manner to $\mathbf{L}$, a square of ionospheric electron density $\mathbf{D}(x, y, z)$ generated by the incident electrons is given by

$$\mathbf{D} = \mathbf{M}_2 \mathbf{f} , \tag{4}$$

where

$$\mathbf{M}_2 = \begin{pmatrix} \mathbf{m}_2 & 0 & \cdots & 0 \\ 0 & \mathbf{m}_2 & \ddots & \vdots \\ \vdots & \ddots & \ddots & 0 \\ 0 & \cdots & 0 & \mathbf{m}_2 \end{pmatrix} . \tag{5}$$

$\mathbf{M}_2$ has the same size as $\mathbf{M}_1$. In (5), the electron density measured by the EISCAT_3D radar was assumed to be caused by the auroral precipitation only. For the derivation of $\mathbf{m}_2$, we assumed that the ionospheric electron density is quasi-steady-state and the ionization loss in the E-layer is dominated by the recombination process. Again, refer to the Appendix of Tanaka et al. (2011) for more detail.

A gray level $g_i$ at a pixel $i$ in the auroral image is approximated by a linear integration along a line of sight, as shown

below:

$$g_i = \frac{c_g(\theta, \phi)}{4\pi} \int L(r, \theta, \phi) dr , \tag{6}$$

where $(r, \theta, \phi)$ are polar coordinates whose origin is located at the center of the camera lens, and $c_g(\theta, \phi)$ is a sensitivity and vignetting factor. Equation (6) can be represented by

$$\mathbf{g} = \mathbf{P}_1 \mathbf{L} = \mathbf{P}_1 \mathbf{M}_1 \mathbf{f} , \tag{7}$$

where $\mathbf{g}$ is a gray-level vector which has $l_g$ elements and $\mathbf{P}_1$ is a $l_g \times m$ matrix used to calculate $\mathbf{g}$ by integrating $\mathbf{L}$ along the line of sight.

The square of electron density observed by the EISCAT_3D radar $\mathbf{d}$ is expressed by the matrix expression:

$$\mathbf{d} = \mathbf{P}_2 \mathbf{D} = \mathbf{P}_2 \mathbf{M}_2 \mathbf{f} , \tag{8}$$

where $\mathbf{P}_2$ is a $l_d \times m$ matrix that extracts data in the voxels corresponding to the radar observation locations from

$\mathbf{D}(x, y, z)$.

Figure 4a shows the modelled ionospheric electron density that should be obtained by the EISCAT_3D radar. In this paper, Gaussian noise with a standard deviation of 5% of the electron density was added to the data. The images taken at five ALIS





stations are presented in Figure 4b. The offset noise of 300 R and the standard deviation of $\sqrt{\mathbf{g}+300}$ R were added to the image data. The finally obtained modelled data are shown as $\tilde{\mathbf{d}}$ and $\tilde{\mathbf{g}}$.

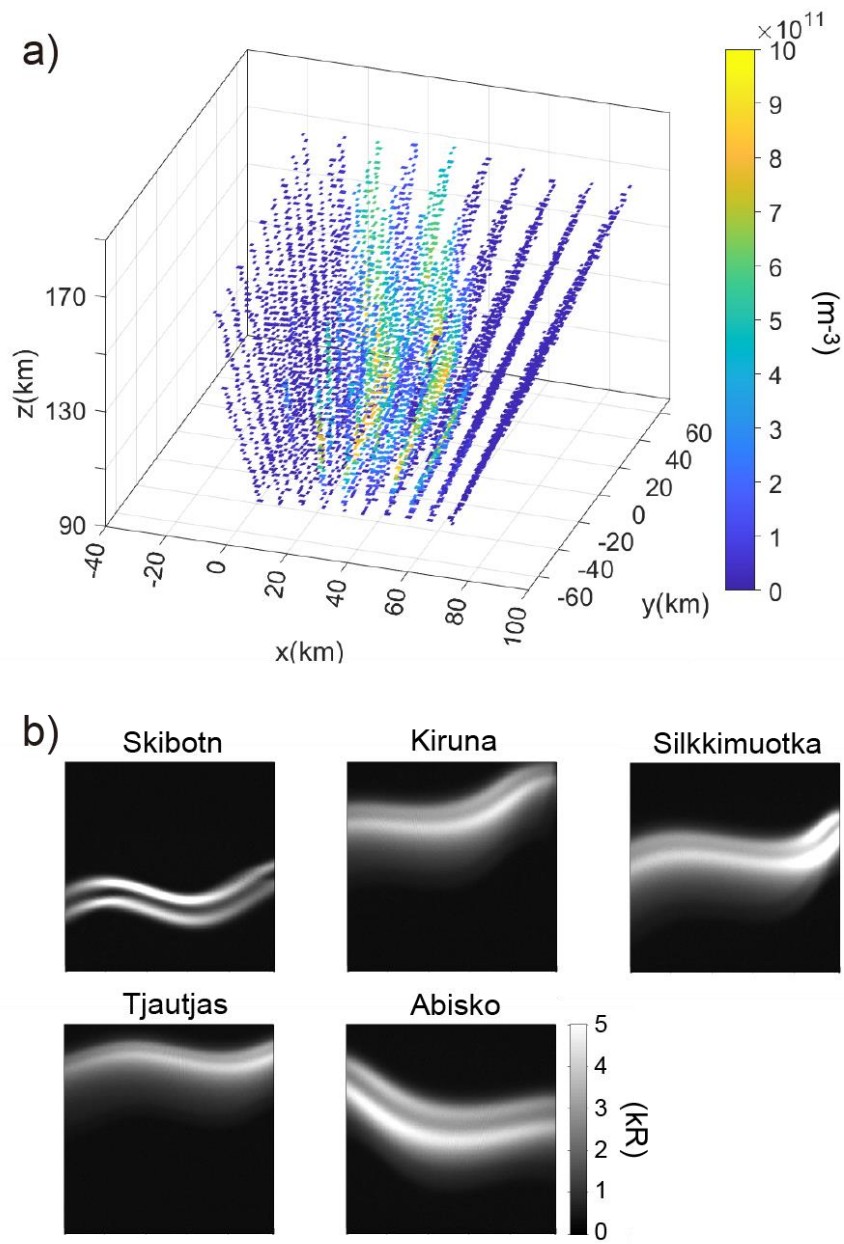

**Figure 4: (a) Modelled ionospheric electron density data obtained by the EISCAT_3D radar. (b) Modelled auroral images taken at five ALIS stations. Top and right of the images correspond to the northward and westward directions, respectively.**






## 3 Inverse analysis

Our inverse analysis method is based on the Bayesian model. According to the Bayes' theorem, the probability that model $\mathbf{f}$

is true after data $\tilde{\mathbf{b}}$ was observed, i.e., the posterior probability $P(\mathbf{f} \mid \tilde{\mathbf{b}})$ is expressed by

$$P(\mathbf{f} \mid \tilde{\mathbf{b}}) = \frac{P(\tilde{\mathbf{b}} \mid \mathbf{f})P(\mathbf{f})}{P(\tilde{\mathbf{b}})} \propto P(\tilde{\mathbf{b}} \mid \mathbf{f})P(\mathbf{f}) , \tag{9}$$

where $P(\tilde{\mathbf{b}} \mid \mathbf{f})$ is the likelihood, which is the probability of observing data $\tilde{\mathbf{b}}$ given model $\mathbf{f}$ , $P(\mathbf{f})$ is the prior

probability of model $\mathbf{f}$ , and $P(\tilde{\mathbf{b}})$ is the marginal probability of $\tilde{\mathbf{b}}$ . In this study, $P(\mathbf{f})$ and $P(\tilde{\mathbf{b}} \mid \mathbf{f})$ are given by

$$P(\mathbf{f}) \propto \exp\left( -\frac{\left\| \nabla^2 \mathbf{f} \right\|^2}{2\sigma^2} \right), \tag{10}$$

$$P(\tilde{\mathbf{b}} \mid \mathbf{f}) \propto \exp\left\{ -\sum_j \frac{1}{2} \left( \tilde{\mathbf{b}}_j - \mathbf{b}_j(\mathbf{f}) \right)^T \Sigma_j^{-1} \left( \tilde{\mathbf{b}}_j - \mathbf{b}_j(\mathbf{f}) \right) \right\} \tag{11}$$

where $\sigma^2$ is the variance of $\nabla^2 \mathbf{f}$ , $\Sigma_j^{-1}$ is the inverse covariance matrix, and $j$ means the kind of data. $\tilde{\mathbf{b}}_j$ corresponds to

the modelled data $\tilde{\mathbf{g}}$ and $\tilde{\mathbf{d}}$ for j = 1 and 2, respectively, and they include the noise. $\mathbf{b}_j(\mathbf{f})$ corresponds to $\mathbf{g}$ and $\mathbf{d}$ in (7)

and (8). Equation (10) indicates the smoothness constraint on $\mathbf{f}$ with respect to $x$, $y$, and $E$. In (11), it was assumed that the

modelled data $\tilde{\mathbf{b}}$ has Gaussian errors. By substituting (10) and (11) into (9), $P(\mathbf{f} \mid \tilde{\mathbf{b}})$ is given by

$$P(\mathbf{f} \mid \tilde{\mathbf{b}}) \propto \exp\left[ -\frac{1}{2\sigma^2} \left\{ \sum_j w_j^2 \left( \tilde{\mathbf{b}}_j - \mathbf{b}_j(\mathbf{f}) \right)^T \Sigma_j^{-1} \left( \tilde{\mathbf{b}}_j - \mathbf{b}_j(\mathbf{f}) \right) + \left\| \nabla^2 \mathbf{f} \right\|^2 \right\} \right] \tag{12}$$

where $w_j$ is a hyper-parameter, which is a constant corresponding to the weighting factor for each instrument data.

Maximization of the posterior probability is equivalent to minimization of the function inside the curly brackets of (12), as

shown below:

$$\varphi(\mathbf{f}; w_j) = \left\| \mathbf{r}(\mathbf{f}; w_j) \right\|^2 , \tag{13}$$

where



$$\mathbf{r}(\mathbf{f};w_1,w_2) = \begin{pmatrix} w_1\boldsymbol{\Sigma}_1^{-\frac{1}{2}}(\tilde{\mathbf{g}}-\mathbf{g}(\mathbf{f})) \\ w_2\boldsymbol{\Sigma}_2^{-\frac{1}{2}}(\tilde{\mathbf{d}}-\mathbf{d}(\mathbf{f})) \\ \nabla^2\mathbf{f} \end{pmatrix}. \tag{14}$$

Here, we change the variables by letting $\mathbf{f} = \exp(\mathbf{x})$ to take advantage of the non-negative constraint of $\mathbf{f}$ (i.e., $\mathbf{f} \geq \mathbf{0}$).

Then, the minimization of $\varphi(\mathbf{x};w_j)$ becomes a non-linear least squares problem with respect to $\mathbf{x}$, so we solved it by the

Gauss-Newton algorithm.

In the Gauss-Newton method, the parameter $\mathbf{x}$ proceeds by the iteration, $\mathbf{x}^{(k+1)} = \mathbf{x}^{(k)} + \Delta\mathbf{x}^{(k)}$, where the increment

$\Delta\mathbf{x}^{(k)}$ at the $k^{\text{th}}$ step is a solution of the following equation:

$$\left(\mathbf{J}^T(\mathbf{x}^{(k)})\mathbf{J}(\mathbf{x}^{(k)})\right)\Delta\mathbf{x}^{(k)} = -\mathbf{J}^T(\mathbf{x}^{(k)})\mathbf{r}(\mathbf{x}^{(k)}), \tag{15}$$

where $\mathbf{J}(\mathbf{x})$ is the Jacobian matrix of $\mathbf{r}(\mathbf{x})$ with respect to $\mathbf{x}$. Since Eq. (15) is a normal equation with a large sparse

matrix, we solved it by the Conjugate Gradient (CG) method. The initial values $\mathbf{x}^{(0)}$ was obtained in advance from only

gray level data $\tilde{\mathbf{g}}$ by solving $\min[\varphi(\mathbf{f})]$ with respect to $\mathbf{f}$. We solved the linear least squares problem (i.e., $\min[\varphi(\mathbf{f})]$)

by the Simultaneous Iterative Reconstruction Technique (SIRT) method (Aso et al., 1998) with $\mathbf{f}^{(0)} = 10^7$ [m$^{-2}$ s$^{-1}$ eV$^{-1}$]

and used the solution $\mathbf{f}*$ for the initial value of the Gauss-Newton algorithm (i.e., $\mathbf{x}^{(0)} = \log(\mathbf{f}*)$). The hyper-parameters

($w_1$, $w_2$) were determined by using the 5-fold cross-validation (Stone, 1974).

## 4 Results from the inverse analysis

The inverse analysis was performed for the reconstruction region shown in Figure 3a (-40 km $< x <$ 100 km, -70 km $< y <$ 70

km) by using the same spatial and energy grids as those for the forward analysis. Figure 5a shows the precipitating electrons'

total energy flux ($Q_0$). Figure 5b indicates $Q_0$ as reconstructed only from five auroral images by the ACT method. In this

paper, the ACT method was used to solve the minimization problem (min[ $\varphi(\mathbf{x};w_j)$ ]) with only the ALIS images. The

results are displayed for the region of -20 km $< x <$ 80 km and -50 km $< y <$ 50 km. It appears that $Q_0$ was reconstructed

well; however, there are two points to be noted: one is an underestimation of $Q_0$ at the peak location of each discrete arc and

the other is an artifact between the two arcs. The energy flux at the center of the reconstructed arcs is slightly smaller than

the input flux. However, the energy flux exists significantly between the two arcs. Figure 5c shows $Q_0$ as reconstructed by

the G-ACT using both the ALIS images and the electron density data from the 10×10 beams of the EISCAT_3D radar. In

this panel, the underestimation of $Q_0$ at the center of each arc and the artifact between the two arcs brought by the normal



ACT were significantly improved. To more clearly show the impact of the electron density data on the improvement, we tested the case that the radar scanned the same area with 21×21 beams. Figure 5d presents $Q_0$ reconstructed by the G-ACT using the electron density from the 21×21 beams. It is evident that $Q_0$ reconstructed by the G-ACT is better improved than that reconstructed by the ACT. Figure 5e shows $Q_0$ derived from only the electron density data from the 21×21 beams of the EISCAT_3D radar. A larger spatial grid size ($\Delta_x = \Delta_y = 5$ km, $\Delta_z = 3$ km) was used for this inverse analysis. Since the

spatial distribution of the electron density data from the radar was much sparser than the number of grids for the ACT and G-ACT cases (i.e., (100, 50, 50) for (x, y, z)), a larger grid size was required to collect enough electron density data to solve the inverse problem, even for the 21×21 beam scan. The two discrete arcs were roughly reconstructed; however, the horizontal resolution was too low to resolve the fine-scale structure of the arcs.





**Figure 5: (a) Horizontal distribution of the incident auroral electrons' total energy flux ($Q_0$). Points A ([x, y]=[35 km, -22 km]), B ([x, y]=[23 km, 16 km]), C ([x, y]=[52 km, -8 km]), and D ([x, y]=[38 km, 16 km]) indicate the locations where the reconstructed height profiles of the electron density and energy spectra of the incident electrons are shown in this paper. (b) $Q_0$ reconstructed from five ALIS images using the ACT method. (c) $Q_0$ reconstructed by the G-ACT method using five ALIS images and the electron density from 10×10 beams of the EISCAT_3D radar. (d) $Q_0$ reconstructed by the G-ACT method using five ALIS images**
**and the electron density from 21×21 beams of the EISCAT_3D radar. (e) $Q_0$ reconstructed only from the electron density from 21×21 beams of the EISCAT_3D radar.**

Figure 6 shows height profiles of the electron density along the field lines at the locations (A, B, C, and D) shown in Figure

5a. These locations were selected to emphasize the difference in the reconstruction between the normal ACT and G-ACT

methods. The black line represents the true profile of the electron density, which was derived by the forward analysis using

the incident electrons described in Section 2.2. The red squares show the modelled electron density data from the 10×10

beams of the EISCAT_3D radar. Several radar data exist along the field line at these locations. It is difficult to estimate the

energy distribution of the precipitating electrons as well as the height distribution of the electron density from such a few

data; this is why the large grid size was used for the inversion from the EISCAT_3D radar data (Figure 5e). The green

crosses and blue circles correspond to the electron density reconstructed by the normal ACT and G-ACT methods,

respectively. The spatial distribution of the electron density data obtained from only the optical images by the ACT is much

denser than those from the EISCAT_3D radar. However, the electron density is smaller than the true values, significantly

above the height of the peak density, which is consistent with the underestimation of $Q_0$ (Figure 5b). The underestimation of

the electron density was significantly modified by the G-ACT using the EISCAT_3D data. What is most important here is

that the electron density can be interpolated at a much higher spatial resolution than that expected from only the

EISCAT_3D radar.



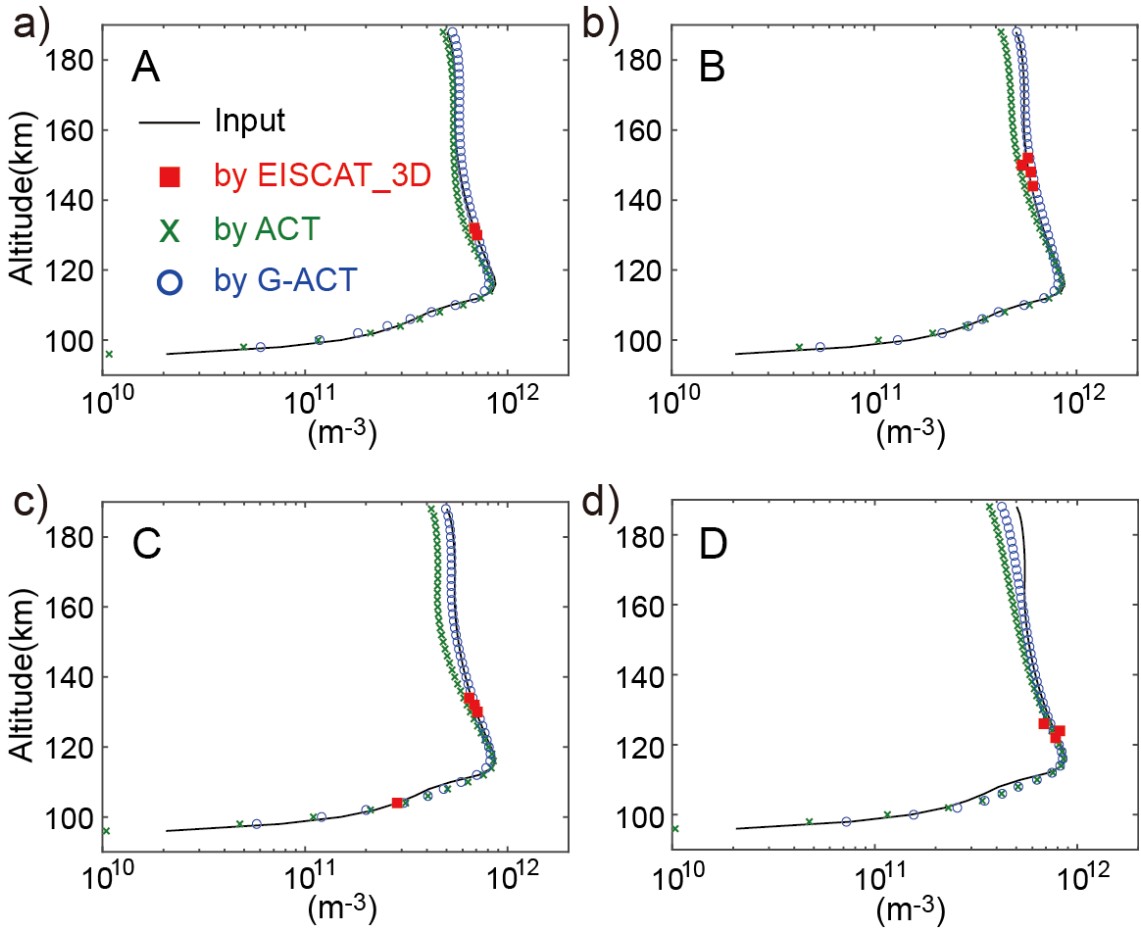

**Figure 6: Height profile of the ionospheric electron density at the points (a) A, (b) B, (c) C, and (d) D, which are shown in Figure 5a. The black curve and solid red squares represent the electron density derived from the incident auroral electrons and modelled observational data from the EISCAT_3D radar (10×10 beams), respectively. Green crosses and blue circles show the electron density reconstructed by the ACT and G-ACT methods, respectively.**

Figure 7 shows the energy distribution of the differential number flux of precipitating electrons at A, B, C, and D, as reconstructed by the ACT and G-ACT methods. The modelled electron density data from the 10×10 beams was used for the G-ACT analysis. This figure indicates that the differential flux reconstructed by the normal ACT tends to be underestimated in the energy range lower than the peak energy ($E_0$). The underestimation of the differential flux was modified by the G-ACT, particularly at the energy corresponding to the altitude where the electron density was obtained. In the assumed situation, the reconstruction results from the G-ACT tends to be better also at the other points (except for A, B, C, and D) than those from the ACT.





**Figure 7: Energy distribution of the differential number flux of the incident electrons at points (a) A, (b) B, (c) C, and (d) D, which are shown in Figure 5a. The black curve shows the energy distribution of the incident electrons. Green crosses and blue circles show the differential number flux reconstructed using the ACT and G-ACT methods, respectively.**

Figures 8a and 8b show $Q_0$ as reconstructed by the ACT and G-ACT methods using data from three ALIS stations (Kiruna, Silkkimuotka, and Tjautjas). All three stations are located to the south of the discrete arcs. Under this condition, it was difficult for the ACT to reconstruct the neighboring multiple arcs precisely from the images because they overlap and cannot





be distinguished from each other. However, it was demonstrated that the G-ACT method using the electron density from the radar is capable of reconstructing the $Q_0$ of multiple arcs. The underestimation of $Q_0$ for both of the two arcs was greatly

improved, although it still remained because the radar beams were somewhat sparse. Again, we tested the 21×21 beam scan case on a trial basis. The reconstructed $Q_0$ was better improved for the both two arcs (Figure 8c). Figures 8d and 8e show the reconstructions made by the ACT and G-ACT when using data from two ALIS stations (Kiruna and Silkkimuotka). In this case, the ACT was not able to separate the two discrete arcs and the northern arc disappeared. The northern arc was partially reconstructed by the G-ACT method, however, the reconstruction was still difficult in the 10×10 beam scan case. Even in

such a case, if a sufficient number of electron density data were available, the G-ACT method was able to reconstruct the $Q_0$ of the two arcs very well (Figure 8f).

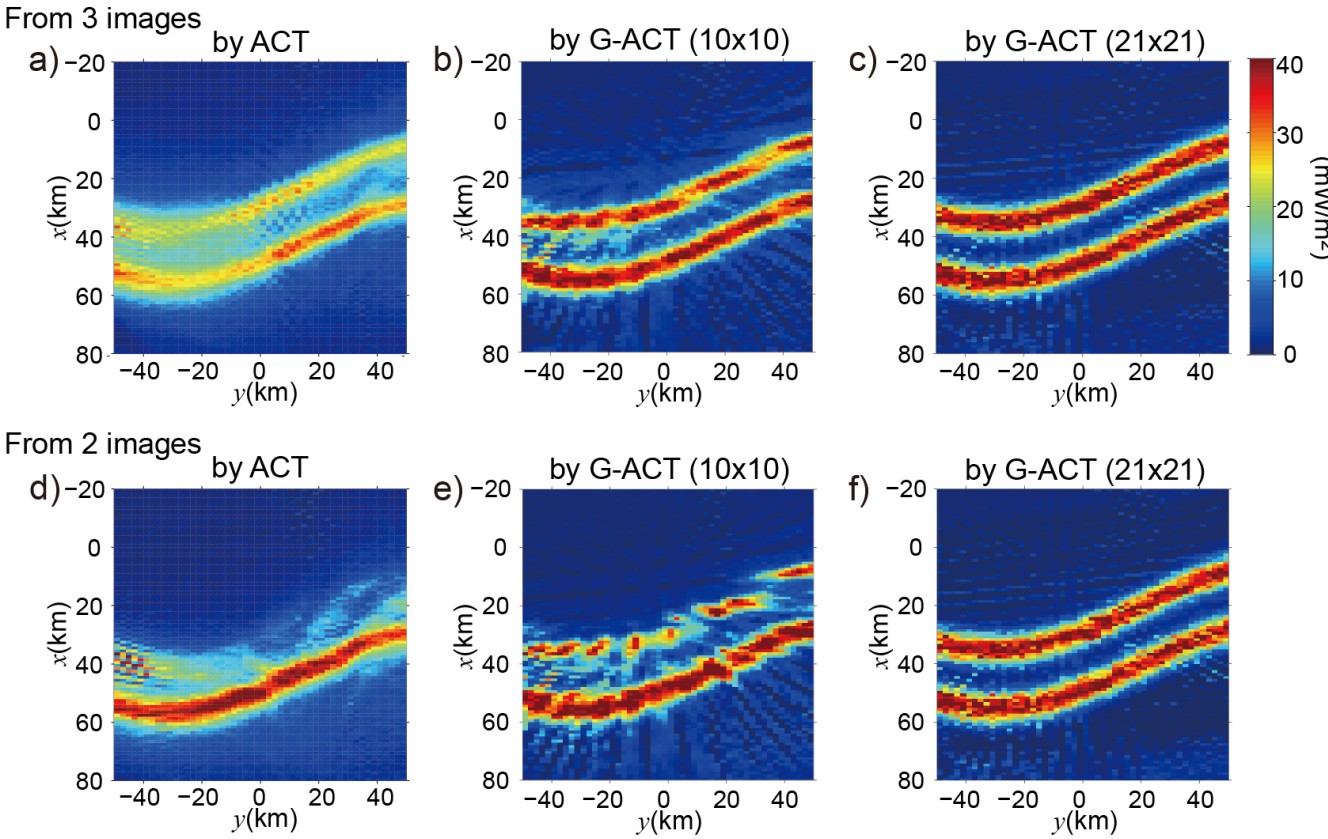

**Figure 8: Upper panels show the total energy flux ($Q_0$) of the incident electrons reconstructed by using three ALIS images (Kiruna, Silkkimuotka, and Tjautjas). (a) $Q_0$ reconstructed by the normal ACT, (b) by the G-ACT using the electron density data from**
**10×10 beams, and (c) by the G-ACT using the electron density data from 21×21 beams. Lower panels show $Q_0$ reconstructed by using two ALIS images (Kiruna and Silkkimuotka). (d), (e), and (f) were obtained by the same method as (a), (b), and (c), respectively.**

Figure 9 shows the height profiles of the electron density at A and B. The upper and lower panels show the reconstructed
results obtained by using three and two ALIS stations, respectively. It can be confirmed that the electron density data




obtained from the EISCAT_3D radar was effectively used to improve the reconstruction result by the normal ACT, particularly around the altitude where the radar data exists. Although the electron density reconstructed by the ACT with a few images is much lower than the true value, the G-ACT enables the underestimation to be corrected.

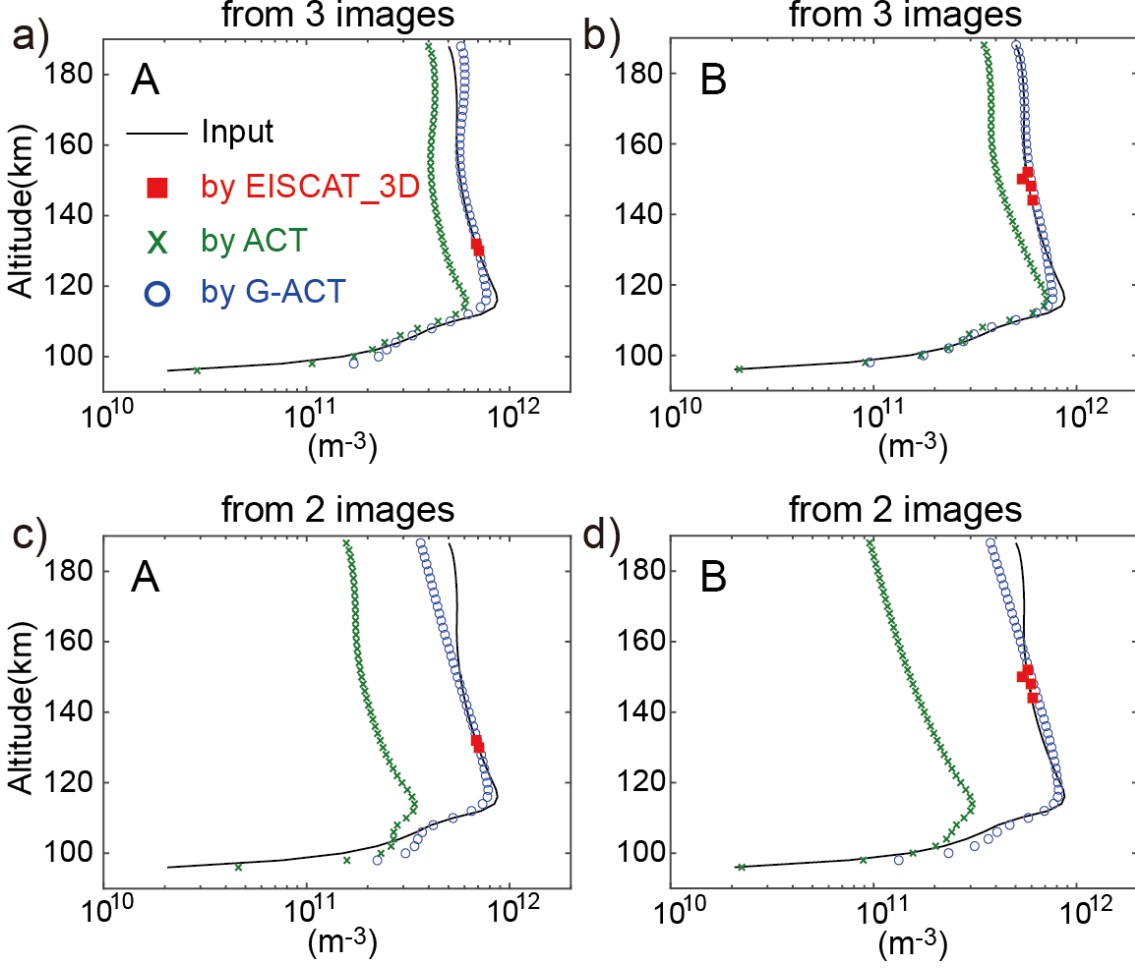


**Figure 9: Height profile of the ionospheric electron density. The format of this figure is similar to that of Figure 6. (a) and (b) indicate the electron density reconstructed by using three ALIS images (Kiruna, Silkkimuotka, and Tjautjas) at A and B, respectively. (c) and (d) show the electron density reconstructed by using two ALIS images (Kiruna and Silkkimuotka) at A and B.**

**5 Discussion**

The EISCAT_3D radar can observe the ionospheric parameters at a much higher spatiotemporal resolution than the existing EISCAT radar. However, if one is interested in the auroral phenomena that have a horizontal scale larger than several tens of kilometers (such as growth-phase arcs, multiple arcs, spirals, westward traveling surges, and omega bands), the spatial distribution of the ionospheric electron density data obtained by the beam scan of the EISCAT_3D radar may be too sparse



to study the fine-scale structures inside the auroras. It is evident that the horizontal spatial resolution is too low to capture
both the entire structure and fine-scale structure of the aurora (e.g., Figures 5e).

The G-ACT method that combines the electron density data with the optical images may enable us to interpolate the electron
density data at a much higher spatial resolution than that observed by the EISCAT_3D radar. In particular, this method is
effective for the reconstruction of the 3D fine-scale structure of an aurora over a wide horizontal area at high temporal
resolution. For instance, this method can provide the fine horizontal structure of the height-integrated ionospheric
conductivity of mesoscale (10–1000 km) auroral phenomena at short sampling intervals. This indicates that it is possible to
estimate 3D current system of such auroral phenomena by using the magnetic field measured by a ground-based
magnetometer network or the ionospheric electric field from radars (e.g., Vanhamäki and Amm, 2007).

Since the auroral images usually include observational noise, it is often difficult to reconstruct the auroral 3D distribution
precisely by using the ACT method, as shown in this study. When the multiple arcs appeared to overlap from all imagers, it
was actually quite difficult for the ACT to distinguish them from each other (Figures 8 and 9). We demonstrated that the G-
ACT can significantly reduce the reconstruction errors caused by the ACT.

Here, we discuss the timescale of auroral phenomena to which the G-ACT method is applicable. We estimated the
integration time required for observing the ionospheric electron density with the EISCAT_3D radar by Eq. (63), (65) and
(66) of Virtanen (2011). It was assumed that the range resolution is 2 km, the full width at half maximum (FWHM) of the
beam is 1.4 degrees, the observation frequency is 233 MHz, and the transmitter power is 3.5 MW, which corresponds to the
power in the first stage of the EISCAT_3D radar. Then, the integration time needed to achieve the standard deviation of 5 %
is less than 0.05 s per one beam in the altitude range of 90-200 km when the electron density is greater than $1.0 \times 10^{11}$ m$^{-3}$.
Thus, the 10×10 beam scan assumed in this study takes about 5 s.

The interpulse period (IPP) between the pulses of the EISCAT_3D radar is about 5 ms for the E-region ionosphere
observation. Thus, if a 16-bit pulse code is used, the minimum temporal resolution becomes 0.08 s per one beam, resulting in
8 s for the 10×10 beam scan. In practice, the temporal resolution depends on the pulse code and background electron density,
therefore, the 10×10 beam scan of the electron density in the E-region ionosphere may be made in less than 5 s.

In addition, the steady state of the electron density was assumed in this study, as given by

$$\frac{\partial N_e}{\partial t} = q - \alpha N_e^2 \approx 0 \,, \qquad (16)$$

in the E-region ionosphere. Here, $q$ is the ion production rate, $N_e$ is the electron density, and $\alpha$ is the effective
recombination coefficient. The steady state condition is satisfied when the incident electron precipitation does not change
over timescales shorter than the ion recombination time constant, $\tau = 1/\alpha N_e$ (e.g., Semeter and Kamalabadi, 2005). It is
well known that $\alpha$ has a large uncertainty (Penman et al., 1979). By using $\alpha$ used by Semeter and Kamalabadi (2005). $\tau$
is between 16 s and 50 s in the altitude of 90-190 km when the electron density is $1.0 \times 10^{11}$ m$^{-3}$, and $\tau$ decreases as
increasing the electron density. Thus, the reconstruction results by the G-ACT using the current model are valid if the auroral

arcs are stable for longer time than $\tau$. However, it is straightforward to add the time derivative term of $N_e$ to our model (i.e.,

$$\tilde{\mathbf{d}} = N_e^2 + (1/\alpha)(\partial N_e / \partial t))$$ because this term can be estimated by the continuous observation of the electron density. Such a modified model is available if $\partial N_e / \partial t$ is stable during the data acquisition interval. We will examine the modification of the model in the near future.

The mesoscale auroras that we mentioned here have the following typical drift speed; 70-170 m/s for the equatorward drift of the growth-phase arcs (Karlsson et al., 2020), 1-2 km/s for the westward traveling surges (Kamide and Baumjohann, 1993), and 200-800 m/s for the eastward drift of the omega bands (Vokhmyanin et al., 2021) at the ionospheric altitude. Pulsating auroras are also mesoscale diffuse aurora, which switch on and off with a quasiperiodic oscillation period of 2-20 s (Lessard, 2012). To study auroral phenomena with relatively fast temporal variations, the number and direction of the beams

need to be adjusted. In such situations, simulation studies as shown in this study may be useful in planning observations.

## 6 Conclusions

We demonstrated via numerical simulation that the combination of optical imagers and the EISCAT_3D radar is very powerful for the study of aurora physics, since they have a complementary relationship with each other. The G-ACT, which was used to reconstruct the 3D distribution of auroras (corresponding to the horizontal 2D distribution of the electron energy

spectra) from multi-instrument data, can be applied as a technique to take advantage of the optical image data effectively. It has the capability to interpolate the electron density observed by the EISCAT_3D radar at a higher spatial resolution, in particular, for mesoscale (10–1000 km) auroral phenomena. The G-ACT enables auroral phenomena to be better reconstructed than when the normal ACT is used. Even if the ACT cannot reconstruct the auroral distribution precisely, the G-ACT may allow us to reduce the reconstruction error. Therefore, it is important to construct a multi-point monochromatic

imager network, including the ALIS, that utilizes the EISCAT_3D radar system in the near future.

## Data availability

The data used in this study are available at ***. (The data used in this study will be prepared for open access when the manuscript is accepted.)

## Author contribution

YT conducted numerical simulation and prepared the manuscript with contributions from all co-authors. YO provided the modeled ionospheric electron density data from the EISCAT_3D radar and contributed to the discussion. AK. TA, BG, UB, TS, GU, and SS contributed to the discussion and interpretation of the simulation results.



**Competing interests**

The authors declare that they have no conflict of interest.

**Acknowledgements**

This study was supported by JSPS KAKENHI Grant Numbers JP17K05672, JP21H01152, and JP22H00173. This work was supported in part by the Inter-university Upper atmosphere Global Observation NETwork (IUGEONET) project (http://www.iugonet.org/). EISCAT is an international association supported by research organizations in China (CRIRP), Finland (SA), Japan (NIPR), Norway (NFR), Sweden (VR), and the United Kingdom (UKRI). ALIS is supported by the
Swedish Research Council. The production of this paper was supported by an NIPR publication subsidy.

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
