# Peer review of "Application of Generalized - Aurora Computed Tomography to the EISCAT\_3D project"

_Annales Geophysicae, 2023_

## Referee Comment (RC2)

**Review of manuscript "Application of Generalized – Aurora Computed Tomography to the EISCAT_3D project" submitted by Tanaka et al. to Ann. Geophys.**

This paper investigates the use of the Generalised Aurora Computed Tomography (G-ACT) method applied to auroral image data from the ALIS network and ionospheric electron densities measured by the upcoming EISCAT_3D radar. The results are compared to similar tomographic methods applied to only one of the two instruments. The results show that G-ACT can reconstruct the volumetric electron density, and precipitation characteristics, at higher resolution than using either instrument alone. This work is important to make efficient use of EISCAT_3D as soon as data becomes available.

Overall the paper is clear and well written, and I don't have any major comments. The main area where I think the paper could be improved is in the explanation of the G-ACT method. Although I realise this is already published in earlier papers, I think it would help readers to make the explanation a little bit clearer. I do not believe this will require any major changes, but just some improvements to the language. A figure or diagram graphically illustrating the steps of the method would also help I think, if possible. I've also listed some minor comments below, mainly minor language corrections, although there are probably other very minor things I've missed which can be fixed in copy editing.

**Moderate comments**

I follow most of the method, but am not sure I have a correct understanding of the difference between **d~** and **d**. At the end of 2.3 it says the finally obtained modelled data are shown as **d~** and **g~**. Are they the same as **d** and **g**, but with noise added? If so this could be more explicitly stated.

The description of the inversion from line 187 onwards could be explained more clearly. I think the main issue is the words "as shown below:" on line 187 – does that refer to equation 13, or all of section 3 after line 187? I suggest adding some words before equation 13 to explain what the equation is for, and/or reordering the description. Perhaps you could add a flow chart showing all of the steps for maximising equation 12, to help the reader to understand?

The study found that ACT underestimates the electron density and electron flux in this case. Could you comment in the discussion on why that might be? Is it expected to be a common situation, or would the density/flux be overestimated just as often?

**Minor comments**

Is there a reason why you use x for the approximately north-south direction and y for the approximately east-west direction? It would be natural to me to name the directions the other way around, and then the axes in figure 3a would be more conventional.

Line 40 is missing a closing parenthesis ).

Line 48/49: "*It* can measure… even though *they* can detect…" - This sentence mixes singular "it" (an optical imager) with plural "they", so needs fixing.

Line 62: I believe the set of authors is not identical between the submitted paper and the papers cited on this line, and for clarity I suggest rewording to remove the use of "we", e.g. "Aso et al., 2008 and Tanaka et al., 2011 extended the ACT method to generalized ACT (G-ACT)."

Line 81: Here (and other places) "field of views" should be "fields of view". Also "each instrument's field of views" should be "the instruments' fields of view" (other wordings possible but the current wording is not quite correct).

Line 93: missing unit "pixels" after 256 x 256.

Line 105: "size" should be plural "sizes".

Line 168: I think you are adding Gaussian noise to the auroral images, but I don't think this is explicitly stated, and should be.

Figure 5: Did you try plotting this with a log scale for the color axis? It might show the electron flux between the arcs produced by the ACT method more clearly.

Line 217: "better improved" could be "more accurate".

Line 241: I suggest "especially" instead of "significantly" (last word of the line).

Line 271: "the both two arcs" should just be "both arcs".

Line 318: Could you comment on how the radar temporal resolution (scan time over all beams) compares to the optical exposure time? Is it relevant?

Line 330: "increasing electron density" should be "the electron density increases".

---

## Author Comment (AC1)

We thank this referee for reviewing our manuscript and providing us the valuable comments. We reply to the referee's comments as follows and have made appropriate changes to the manuscript.

*The manuscript demonstrates that it should be possible to use generalized aurora computed tomography to reconstruction the electron density over an area of the ionosphere using multiple simultaneous optical images, thereby extending the radar observations thereof. The manuscript is generally well-written and comprehensive. The paper is nearly ready for publication with a few minor revisions described below:*

**Comment 1**
*L211: "artifact" should be "artefact". More importantly, the non-expert eye cannot know where the artefact in Figure 5 is. Please describe it or point it out clearly.*

**Reply 1:**
We modified "artifact" to "overestimation" in the manuscript. Furthermore, we added the following description to the paragraph from L205.

It appears that $Q_0$ was reconstructed well; however, there are two points to be noted: one is an underestimation of $Q_0$ at the peak location of each discrete arc and the other is an overestimation between the two arcs. The energy flux at the center of the reconstructed arcs is slightly smaller than the input flux. On the other hand, the energy flux between the two arcs is greater than the input flux, particularly at $y < 0$. For example, $Q_0$ at $(x, y)$=(45km, -20km) is 1.47 mW/m$^2$ for the input flux and 7.30 mW/m$^2$ for the reconstructed one by the ACT. The $Q_0$ at the location was improved to 4.36 mW/m$^2$ (2.29 mW/m$^2$) by the G-ACT method with the electron density from 10×10 beams (21×21 beams) of the EISCAT_3D radar.

**Comment 2**
*L215: "significantly improved", L217: "better improved", L269: "greatly improved", L271: "better improved", L288: "much lower" are all subjective statements. "significantly", "better", "greatly" and "much" mean different things to different readers. The authors should quantify what they mean by these subjective descriptors please.*

**Reply 2:**
To quantify the performance of the reconstruction methods used in this paper, we calculated the Mean Absolute Error (MAE) or the Mean Absolute Percentage Error (MAPE). The MAE was used for the total energy flux (Figure 5 and 8) because the total

energy flux includes values close to zero, whereas the MAPE was used for the electron density (Figure 6 and 9) and the differential number flux of the precipitating electrons (Figure 7) because they have a wide scale (e.g., from $10^7$ to $10^{10}$ s$^{-1}$m$^{-2}$eV$^{-1}$ for the differential number flux).

The MAE and MAPE are defined by

$$MAE = \frac{1}{N}\sum_i^N |\hat{y}_i - y_i|,$$

$$MAPE = \frac{1}{N}\sum_i^N \left|\frac{\hat{y}_i - y_i}{y_i}\right| \times 100,$$

respectively, where $\hat{y}_i$ is the reconstruction and $y_i$ the true (input) value.

The MAE for the total energy flux was calculated by using all data in the evaluation area (-20 km $< x <$ 80 km, -50 km $< y <$ 50 km). The MAPE for the electron density and the differential number flux was calculated by using data at the points A, B, C, and D. The MAE and MAPE values are summarized in the following table.

| Figure | Panel | Reconstruction Method | MAE or MAPE | Value |
|--------|-------|----------------------|-------------|-------|
| 5 | b) | ACT | MAE | 2.11 [mW/m$^2$] |
| | c) | G-ACT (10x10 beams) | | 1.87 [mW/m$^2$] |
| | d) | G-ACT (21x21beams) | | 1.68 [mW/m$^2$] |
| 6 | a) | ACT | MAPE | 6.3 % |
| | | G-ACT (10x10 beams) | | 5.4 % |
| | b) | ACT | | 9.9 % |
| | | G-ACT (10x10 beams) | | 3.4 % |
| | c) | ACT | | 13.6 % |
| | | G-ACT (10x10 beams) | | 4.3 % |
| | d) | ACT | | 12.9 % |
| | | G-ACT (10x10 beams) | | 6.4 % |
| 7 | a) | ACT | MAPE | 38.4 % |
| | | G-ACT (10x10 beams) | | 30.1 % |
| | b) | ACT | | 40.1 % |
| | | G-ACT (10x10 beams) | | 35.7 % |
| | c) | ACT | | 50.5 % |
| | | G-ACT (10x10 beams) | | 41.2 % |
| | d) | ACT | | 55.1 % |

| | | | | |
|---|---|---|---|---|
| | | G-ACT (10x10 beams) | | 50.0 % |
| 8 | a) | ACT | MAE | 4.46 [mW/m$^2$] |
| | b) | G-ACT (10x10 beams) | | 3.19 [mW/m$^2$] |
| | c) | G-ACT (21x21beams) | | 1.86 [mW/m$^2$] |
| | d) | ACT | | 5.87 [mW/m$^2$] |
| | e) | G-ACT (10x10 beams) | | 3.97 [mW/m$^2$] |
| | f) | G-ACT (21x21beams) | | 1.80 [mW/m$^2$] |
| 9 | a) | ACT | MAPE | 25.2 % |
| | | G-ACT (10x10 beams) | | 9.5 % |
| | b) | ACT | | 24.2 % |
| | | G-ACT (10x10 beams) | | 4.6 % |
| | c) | ACT | | 62.6 % |
| | | G-ACT (10x10 beams) | | 17.6 % |
| | d) | ACT | | 66.0 % |
| | | G-ACT (10x10 beams) | | 13.6 % |

All the MAE and MAPE values for the G-ACT reconstruction are smaller than those for the ACT reconstruction. We added these values to Figures 6-9 and the description about the MAE and MAPE to the revised manuscript.

---

## Author Comment (AC2)

We thank this referee for reviewing our manuscript and providing us the valuable comments. We reply to the referee's comments as follows and have made appropriate changes to the manuscript.

*Moderate comments*

*Comment 1*

*I follow most of the method, but am not sure I have a correct understanding of the difference between d~ and d. At the end of 2.3 it says the finally obtained modelled data are shown as d~ and g~. Are they the same as d and g, but with noise added? If so this could be more explicitly stated.*

**Reply 1:**

Yes, they are. $\tilde{\mathbf{d}}$ and $\tilde{\mathbf{g}}$ were made by just adding noise to $\mathbf{d}$ and $\mathbf{g}$, respectively. We modified the last paragraph of the section 2.3 to make that clear, as shown below.

We added the noise to $\mathbf{d}$ and $\mathbf{g}$ and finally obtained modelled data, $\tilde{\mathbf{d}}$ and $\tilde{\mathbf{g}}$. Gaussian noise with a standard deviation of 5% of the electron density was added to the electron density data. The offset of 300 R was added to the gray level data and then Gaussian noise with a standard deviation of $\sqrt{\mathbf{g}+300}$ R were added. Figure 4a and 4b show the modelled ionospheric electron density that should be obtained by the EISCAT_3D radar and the modelled auroral images at five ALIS stations.

*Comment 2*

*The description of the inversion from line 187 onwards could be explained more clearly. I think the main issue is the words "as shown below:" on line 187 – does that refer to equation 13, or all of section 3 after line 187? I suggest adding some words before equation 13 to explain what the equation is for, and/or reordering the description. Perhaps you could add a flow chart showing all of the steps for maximising equation 12, to help the reader to understand?*

**Reply 2:**

We revised the description after line 180, as shown below. In addition, we summarized the flow of the inverse analysis at the end of this paragraph.

[revised manuscript text omitted]

***Comment 3***

*The study found that ACT underestimates the electron density and electron flux in this case. Could you comment in the discussion on why that might be? Is it expected to be a common situation, or would the density/flux be overestimated just as often?*

**Reply 3:**

As for the multiple arcs assumed in this study, the total energy flux of the precipitating electrons reconstructed by ACT were underestimated inside the discrete arcs. However, different conditions such as the relative position of the aurora and the imagers, the noise level, and the shape of the aurora cause the reconstructed electron flux to be overestimated. I revised the paragraph from line 308, as shown below.

Since the auroral images usually include observational noise, it is often difficult to reconstruct the auroral 3D distribution precisely by using the ACT method. As for the multiple arcs assumed in this study, the total energy flux of the precipitating electrons reconstructed by ACT was underestimated inside the discrete arcs. This is because that the two neighboring arcs overlapped when viewed from several imagers and were difficult to perfectly separate. In Figure 5b, the reconstructed electron flux between the arcs was greater than the modelled flux, and instead, the flux inside the arcs decreased. When the multiple arcs overlapped from all imagers, it was quite difficult for the ACT to distinguish them from each other (Figure 8). Of course, the reconstruction result depends on the condition such as the relative position of the aurora and the imagers, the noise level, and the shape of the aurora, and different conditions cause the reconstructed electron flux to be overestimated. We demonstrated that the G-ACT can significantly reduce the reconstruction errors caused by the ACT.

*Minor comments*

*Is there a reason why you use x for the approximately north-south direction and y for the approximately east-west direction? It would be natural to me to name the directions the other way around, and then the axes in figure 3a would be more conventional.*

The coordinate system with x for the magnetically north-south direction and y for the east-west direction was used by Tanaka et al. [2010]. Since we followed the analysis

method described in Tanaka et al. [2010], we would like to use the same coordinate system.

*Line 40 is missing a closing parenthesis ).*

We modified it.

*Line 48/49: "It can measure… even though they can detect…" - This sentence mixes singular "it" (an optical imager) with plural "they", so needs fixing.*

We replaced "they" with "it".

*Line 62: I believe the set of authors is not identical between the submitted paper and the papers cited on this line, and for clarity I suggest rewording to remove the use of "we", e.g. "Aso et al., 2008 and Tanaka et al., 2011 extended the ACT method to generalized ACT (G-ACT)."*

We revised this sentence according to the reviewer's suggestion.

*Line 81: Here (and other places) "field of views" should be "fields of view". Also "each instrument's field of views" should be "the instruments' fields of view" (other wordings possible but the current wording is not quite correct).*

We revised them according to the reviewer's comment.

*Line 93: missing unit "pixels" after 256 x 256.*

We added "pixels" after 256 x 256.

*Line 105: "size" should be plural "sizes".*

We modified it.

*Line 168: I think you are adding Gaussian noise to the auroral images, but I don't think this is explicitly stated, and should be.*

We clearly stated that Gaussian noise was added to the auroral images, as shown in Reply 1.

*Figure 5: Did you try plotting this with a log scale for the color axis? It might show the electron flux between the arcs produced by the ACT method more clearly.*

We tried plotting Figure 5 with a log scale for the color axis. However, the plots emphasize too much the region where the total energy flux is small and we do not focus on. Thus, we left Figure 5 as it is and added the following description to the paragraph from L205.

It appears that $Q_0$ was reconstructed well; however, there are two points to be noted: one is an underestimation of $Q_0$ at the peak location of each discrete arc and the other is an overestimation between the two arcs. The energy flux at the center of the reconstructed arcs is slightly smaller than the input flux. On the other hand, the energy flux between the two arcs is greater than the input flux, particularly at $y < 0$. For example, $Q_0$ at $(x, y)$=(45km, -20km) is 1.47 mW/m$^2$ for the input flux and 7.30 mW/m$^2$ for the reconstructed one by the ACT. The $Q_0$ at the location was improved to 4.36 mW/m$^2$ (2.29 mW/m$^2$) by the G-ACT method with the electron density from 10×10 beams (21×21 beams) of the EISCAT_3D radar.

*Line 217: "better improved" could be "more accurate".*

We modified it. In addition, we calculated the Mean Absolute Error (MAE) or the Mean Absolute Percentage Error (MAPE) to quantify the performance of the reconstruction methods and added these values to Figures 5-9 and the text. Please see "Reply on RC1" more in detail.

*Line 241: I suggest "especially" instead of "significantly" (last word of the line).*

We modified it.

*Line 271: "the both two arcs" should just be "both arcs".*

We modified it.

*Line 318: Could you comment on how the radar temporal resolution (scan time over all*

*beams) compares to the optical exposure time? Is it relevant?*

The temporal resolution of optical imager depends on the performance of the imager, the wavelength of the filter, the auroral emission intensity, etc. Since the monochromatic images are required for the G-ACT analysis, the temporal resolution of high-sensitivity imagers (e.g., electron-multiplying CCD (EMCCD) imagers) with the monochromatic filters is usually a few seconds and can be higher than that of the 10×10 beam scan of the radar. For example, Fukizawa et al. (2022) reconstructed the 3D distribution of pulsating aurora every 2 seconds by the ACT using the 427.8-nm auroral images.
We added the above description in Discussion.

*Line 330: "increasing electron density" should be "the electron density increases".*

We modified it.